# Lightweight Substation Equipment Defect Detection Algorithm for Small Targets

**DOI:** 10.3390/s24185914

**Published:** 2024-09-12

**Authors:** Jianqiang Wang, Yiwei Sun, Ying Lin, Ke Zhang

**Affiliations:** 1Department of Electronic and Communication Engineering, North China Electric Power University, Baoding 071003, China; 220222215019@ncepu.edu.cn (J.W.); zhangkeit@ncepu.edu.cn (K.Z.); 2State Grid Shandong Electric Power Research Institute, Jinan 250003, China; sddkysyw@163.com; 3Hebei Key Laboratory of Power Internet of Things Technology, North China Electric Power University, Baoding 071003, China

**Keywords:** defect detection, deep learning, substation equipment, small object detection, lightweight, YOLOv8

## Abstract

Substation equipment defect detection has always played an important role in equipment operation and maintenance. However, the task scenarios of substation equipment defect detection are complex and different. Recent studies have revealed issues such as a significant missed detection rate for small-sized targets and diminished detection precision. At the same time, the current mainstream detection algorithms are highly complex, which is not conducive to deployment on resource-constrained devices. In view of the above problems, a small target and lightweight substation main scene equipment defect detection algorithm is proposed: Efficient Attentional Lightweight-YOLO (EAL-YOLO), which detection accuracy exceeds the current mainstream model, and the number of parameters and floating point operations (FLOPs) are also advantageous. Firstly, the EfficientFormerV2 is used to optimize the model backbone, and the Large Separable Kernel Attention (LSKA) mechanism has been incorporated into the Spatial Pyramid Pooling Fast (SPPF) to enhance the model’s feature extraction capabilities; secondly, a small target neck network Attentional scale Sequence Fusion P2-Neck (ASF2-Neck) is proposed to enhance the model’s ability to detect small target defects; finally, in order to facilitate deployment on resource-constrained devices, a lightweight shared convolution detection head module Lightweight Shared Convolutional Head (LSCHead) is proposed. Experiments show that compared with YOLOv8n, EAL-YOLO has improved its accuracy by 2.93 percentage points, and the mAP50 of 12 types of typical equipment defects has reached 92.26%. Concurrently, the quantity of FLOPs and parameters has diminished by 46.5% and 61.17% respectively, in comparison with YOLOv8s, meeting the needs of substation defect detection.

## 1. Introduction

Substations play a crucial role in the power system [1]. In daily operation and maintenance, substation equipment is complex and numerous, and manual operation and maintenance are stressful. Equipment failure poses a severe challenge to the stable operation of the system [2]. Research on substation defect detection technology is an important topic in the construction of a new generation of intelligent substations. At present, defect detection mainly uses deep learning methods [3] to realize functions such as substation equipment status monitoring, station-side meter defect identification, and personnel safety assessment. Therefore, substation defect detection technology based on artificial intelligence neural network algorithms has great research prospects. Reference [4] made improvements based on the YOLOv5 algorithm. It addressed the problem of large differences in defect features of substation equipment, enhanced the semantic distinguishability of the model, and improved the accuracy of defect detection. Reference [5] improved the Faster R-CNN [6] algorithm and proposed a defect detection algorithm for small target oil stains, which improved the accuracy of oil stain detection. However, single defect detection cannot adapt to complex substation scenarios. Reference [7] improved YOLOv5 by introducing the Transformer [8] structure into the YOLOv5 network and introducing focal loss to optimize small sample data to enhance the precision and resilience of the algorithm. Nonetheless, the Transformer architecture has traditionally grappled with a substantial quantity of parameters, which has led to suboptimal real-time performance of the algorithm and constrained its deployment potential.

From the existing research, substation defect detection algorithms are mostly focused on improving accuracy, and do not comprehensively consider the characteristics of substation scenarios. In the substation defect detection scenario, data collection is mostly carried out through cameras, drones and other equipment. Due to the shooting angle, viewing angle and other reasons, targets of different sizes will be produced, and there will also be small-sized equipment targets, which will cause some defects to exist in such small target equipment. However, small target defect detection is not considered in existing algorithms. In addition, the intricate background information pertaining to the substation can also impact the feature extraction capabilities of the model. Finally, to ensure deployment on resource-constrained devices, the complexity of the algorithm also needs to be considered. In response to the above problems, this paper proposes a small target and lightweight substation main scene equipment defect detection algorithm: EAL-YOLO, which effectively improves the model’s detection capabilities in complex scenes and small target defect scenes. The detection accuracy of typical defect types such as typical defects of meters, typical defects of respirators, and secondary equipment status can reach more than 85%. Compared with the baseline model, the detection accuracy is improved by 2.93% while taking into account the algorithm’s FLOPs and parameters. The main contributions of this paper include:

(1) When confronted with complex scenarios involving substation equipment, this paper uses the lightweight ViT model, EfficientFormerV2 [9] to redesign the backbone in YOLOv8. Furthermore, it introduces the LSKA [10] mechanism within the SPPF layer of the backbone, enhancing the algorithm’s feature extraction capabilities in complex environments.

(2) This paper introduces the ASF2-Neck module, designed specifically for the detection of small target defects. To circumvent the expansion of model parameters and FLOPs typically associated with the large feature maps of diminutive targets, the module incorporates a dedicated branch for small target features within the neck network architecture. Empirical evidence demonstrates that the ASF2-Neck module significantly enhances the model’s precision and its capacity to detect small target defects.

(3) In order to further simplify the model complexity, this paper proposes a LSCHead, which uses a shared convolution design to predict feature outputs through the same detection head and uses Group Normalization (GN) [11] to optimize the shared convolution module. The LSCHead presented in this paper substantially diminishes the complexity of the model while simultaneously enhancing its accuracy.

## 2. Related Work

### 2.1. Substation Equipment Defect Detection

In recent years, a multitude of scholars from around the globe have invested substantial research efforts into the field of substation defect detection. Reference [12] used a method that combines semantic segmentation with target detection, and used the method of segmentation first and then detection to isolate the influence of complex environment on equipment defect detection. However, it only detects one defect. In the face of complex scene equipment in substations, the robustness of its algorithm was limited. Reference [13] improved Faster R-CNN, introduced a new loss function and feature fusion network for substation equipment defect detection, and overcame the interference of complex environments. However, the large amount of FLOPs and parameters of Faster R-CNN made the algorithm difficult to deploy on mobile devices. Reference [14] improved the traditional CNN, improved the anti-interference ability of the detection algorithm in complex environments, and realized the detection of the disconnection state of the disconnector and the positioning of the insulator and switch of the disconnector. Concurrently, the algorithm was not optimized for detecting small targets, resulting in a high rate of missed detections for such objects. 

### 2.2. YOLOv8 Model

The YOLO series, initially proposed by Joseph Redmon in 2016 [15], stands out among other target detection algorithms by its ability to satisfy the demands of real-time detection. Consequently, it has become widely adopted in various industrial projects. YOLOv8 represents the latest iteration of the YOLOv5 series, introduced by Ultralytics in 2023. It enhances capabilities by extending support to tasks including object detection, instance segmentation, and pose prediction. YOLOv8 includes five versions: n, s, m, l, and x. The number of parameters and the amount of FLOPs in different versions increase accordingly. YOLOv8n has a small size and a fast detection rate. It can be deployed on mobile devices such as substation inspection robots and drones while ensuring high accuracy. Considering the excellent performance of YOLOv8, this paper chooses YOLOv8n as the baseline model. The architectural framework of YOLOv8 is depicted in Figure 1.

### 2.3. YOLOv8 Enhancement Research

The Convolutional Neural Network(CNN) is the main technical support for current object detection technology due to its simple structure and high training efficiency. However, it fails to effectively capture the global information of the image; in contrast, ViT’s unique Multi-Head Self-Attention (MHSA) can effectively extract the global information from the image. An existing proposal for enhancement involves integrating ViT as the backbone of YOLOv8 to bolster the model’s feature extraction capabilities. However, it’s important to note that the pure ViT model comes with a significant number of parameters and FLOPs, which cannot be overlooked. The mainstream SwinTransformer [16] and CSwinTransformer [17] rely on their unique window attention mechanism to alleviate the huge number of parameters and FLOPs brought by ViT, but their parameters and FLOPs are still not negligible. The mainstream lightweight networks such as MobleNet [18] and RepVGG [19] can meet the lightweight requirements with their simple architecture of pure convolution, but they will bring a certain decline in model performance. Considering the high precision brought by ViT and the lightweight properties of CNN, hybrid networks based on CNN and ViT have emerged at this stage, such as EfficientViT [20] and EfficientFormer [21]. The emergence of these networks enables the model to improve feature extraction capabilities while taking into account its complexity. Another improvement idea is to improve the neck. Recent studies have shown that improving the network at the neck can improve the performance of the model in a targeted manner [22,23,24,25], but from the existing methods, the neck design for small targets is still insufficient. Among the ideas for improving the head, some mainstream target detectors have already used convolution heads with shared parameters, and the effect is obvious [26,27]. The purpose of the shared convolution head is to process features of different scales and make them at the same feature scale, and then send them to the convolution head with shared parameters. All convolution heads share the same parameters, achieving a significant lightweight effect.

## 3. EAL-YOLO

Aiming at the defect characteristics of equipment in the main scene of substations, as well as its requirements for the deployment of edge devices such as drones, intelligent inspection robots and cameras, this paper enhances the original YOLOv8 and introduces EAL-YOLO. First, EfficientFormerV2 is used as the backbone of the model, and the LSKA mechanism is integrated into the SPPF to bolster the model’s feature extraction capabilities. Secondly, based on the neck architecture in ASF-YOLO [22], the P2 feature layer is incorporated to design the ASF2-Neck network, which improves the model’s small target defect detection capabilities. As shown in the Neck part in Figure 2, the colored modules are the added P2 feature inputs. To further optimize the model’s lightweight architecture, the paper presents a novel lightweight target detection head, designated as LSCHead. It employs the concept of shared parameters to further streamline the model’s complexity. As shown in the Head part in Figure 2, the modules with the same color are all parameter sharing modules.

### 3.1. Analysis and Enhancement of the Backbone

To bolster the model’s feature extraction capabilities without neglecting its complexity, this paper employs a hybrid network architecture that combines CNN and ViT, thereby enhancing the accuracy of substation equipment defect detection. The specific architecture of the backbone utilized in this study is depicted in Figure 3. As can be seen from Figure 3, the entire network architecture is divided into a Local module with CNN as the main body, a Subsample module, and a LocalGlobal module with ViT as the main body. Concurrently, the LSKA mechanism is employed at the network’s final output stage to bolster the acquisition of global information. Among them, Local introduces a 3 × 3 depth-separable convolution as an intermediate layer to realize the functions of pooling and convolution, and also captures local information with a small performance overhead. The LocalGlobal module designed based on the ViT architecture is divided into LightweightLocalGlobal (LLG) which is mainly lightweight and ViT module LocalGlobal (LG) which is mainly used to capture global information. The LG module obtains global information through the *MHSA* mechanism and then outputs it in a fixed size through the linear layer. Among them, the *MHSA* mechanism is shown in Formula (1):(1)$$MHSA(Q,K,V)=Soft\max(Q\cdot K^{T}+ab)\cdot V$$

Since the ViT architecture is only inserted in the back layer of the network, it is impossible to take into account the feature outputs of different sizes. The design of LLG is to overcome this problem. By downsampling all *Q*, *K*, and *V* to a uniform resolution, and then inserting the original resolution into the next layer through the attention layer, this approach brings fewer parameters than the pure ViT architecture, and can also effectively improve performance. Subsample adopts an architecture similar to LLG, but adds the idea of residual to the overall structure, and captures dynamic and static feature information through pooling and convolution. The convolution of large kernels can bring a larger receptive field, which can effectively improve the performance of the model in the face of complex scene information of substations. The main improvement of the SPPF-LSKA module is the LSKA module. Considering the increase in parameters and FLOPs by large kernel convolution, LSKA adopts a separable form, which divides the 7 × 7 large kernel convolution into 3 × 3 DW.Conv and [7/2] × [7/2] DW.D.Conv for local and global information extraction respectively, which is equivalent to directly using 7 × 7 large kernel convolution, while simultaneously reducing the model’s number of parameters and FLOPs. The backbone design of this paper integrates the hybrid network model and the LSKA mechanism enhances the feature extraction capabilities of the defect detection algorithm.

### 3.2. ASF2-Neck Network for Small Target Defects

According to the analysis of the characteristics of the substation dataset, it is found that there are many defects in the size of small targets and the defect information is complex. The feature fusion network within conventional detection models directly merges feature information across various scales through the process of upsampling and downsampling, as shown in the Neck part of Figure 1. This fusion strategy is to fuse different scale features separately, and it is impossible to achieve information interaction between different scales. The design idea of the neck in this paper is to add a P2 feature input layer on the basis of the general neck improvement. In the general backbone feature output, the feature size of the P2 output is 160 × 160. The purpose of introducing this feature output is to detect small targets of 4 × 4 size. Extremely small targets below 4 × 4 are usually defined as invalid points for substation defect detection, so the detection capability of 4 × 4 and above can meet the needs of substation defect detection. The concept for improvement presented in this paper is illustrated in Figure 4.

Compared to the original YOLOv8 neck architecture, ASF2-Neck incorporates the SSFF module and the Zoom_cat module, designed to enhance information exchange among various scales. The detailed architecture of the SSFF module and the Zoom_cat module is depicted in Figure 5. The SSFF module is designed to integrate the features generated by the backbone. It first scales the features of different scales through the Unsqueeze module and the Unsample module, then stacks the features through the Stack as the input of 3D Conv, and finally outputs them through the normalization layer. The Zoom_cat module is designed to acquire detailed information about features at various scales. First, the features of larger scales are processed by the convolution module, then downsampled through the pooling layer, and finally output through the convolution module. At this time, the large-scale features will be processed to be equivalent to the feature scale of the intermediate size. The feature scale of the intermediate size is only processed by the convolution module, while the small-scale features are upsampled by the nearest neighbor interpolation method after passing through the convolution module, so that their feature scale is equivalent to the feature of the intermediate size, and finally the three are concat spliced. The Neck that introduces the Zoom_cat module and the SSFF module can effectively realize the information interaction between features of different scales, but the original ASF-Neck is still limited in detecting small targets. Combined with the idea of Figure 4, this paper introduces the P2 feature output layer using the SSFF module on this basis, so that the model focuses more intently on minor target defects. The newly designed Neck is called ASF2-Neck.

As shown in the ASF2-Neck part of Figure 2, the orange module is the newly added P2 detection layer. In order to reduce the amount of FLOPs and parameters of the model, only ADD is used to add different features, and no redundant steps are included. From ASF2-Neck, it can be seen that on the basis of the original neck, the P2 feature output of the backbone network is directly introduced into the Concat module, and the fused P3, P4, and P5 features are further fused with P2 using the SSFF module. Considering the characteristics of the Zoom_cat module, if the P2 layer is directly introduced into the front end of feature fusion, the large feature map preprocessed by the Stem module will directly participate in the feature fusion, and the amount of parameters brought is huge. Therefore, P2 is placed at the back end of feature fusion and only the SSFF module is used for fusion without considering Zoom_cat. This consideration is beneficial to the lightweight model. Compared with the shortcoming of P2 not participating in feature fusion in the YOLOv8n neck network, the newly designed ASF2-Neck further considers the scenario of small target defect detection on the basis of the original ASF-Neck, and effectively detects small target defects larger than 4 × 4 by taking advantage of the large scale of the P2 feature layer. Admittedly, the introduction of new feature outputs will inevitably increase the complexity of the model. In Section 3.3, this paper proposes a new lightweight detection head to control the parameters of the model within a reasonable range while maintaining the stability of the detection accuracy.

### 3.3. Lightweight Shared Convolution Head (LSCHead)

Substation equipment defect detection algorithms are usually deployed on edge devices such as inspection robots, cameras, and drones. Complex model algorithms that are overly intricate pose challenges when deployed on edge devices with constrained computational resources. At the same time, the original YOLOv8 architecture uses corresponding detection heads for detection of P2-P5 features, which inevitably causes redundancy in computational capacity. Considering the need for lightweight algorithms for substation defect detection, this paper introduces a lightweight detection head, designated as LSCHead, which is constructed upon the foundation of shared convolution. The design of its structure is shown in the LSCHead part in Figure 2. Among them, the colored modules are all shared parameter modules, which greatly reduces the redundancy of parameters. In the original YOLOv8 head design, it uses independent detection heads at different feature layers. Even if the structures of these detection heads are the same, such a design will reduce the utilization of parameters. At the same time, after adding the P2 small target layer, the number of parameters will be further increased. This paper adopts the idea of shared parameters and designs a lightweight shared convolution head LSCHead, which has a simple structure and significant effect, and can substantially decrease the complexity of the model. In the design of LSCHead, a very simple structure is adopted to avoid the increase of model parameters due to complex architecture. At the same time, BatchNormalization (BN) in ConvMoudle is replaced by GroupNormalization (GN) with better performance. BN uses normalization along batch size to enable various networks to be trained, but when batch size is too small, its error will increase significantly [11]. GN can avoid the influence of different batch sizes on its accuracy by grouping channels. Its specific normalization method is shown in Figure 6. The colored modules are the normalization methods of BN and GN.

All normalization operations are based on the same formula; the difference is the dimension in which they are performed. The formula is shown in (2):(2)$${\hat{x}}_{i}=\frac{1}{\sigma_{i}}(x_{i}-\mu_{i})$$
where x represents the feature input, for the input image, i is a four-dimensional vector, $i=(i_{N},i_{C},i_{H},i_{W})$, where N represents the batch, C represents the number of channels, H and W represent the height and width of the feature map. μ and σ represent the mean and standard deviation, and the formula is shown in (3):(3)$$\mu_{i}=\frac{1}{m}\sum_{k\in S_{i}}x_{k},\;\;\;\;\sigma_{i}=\sqrt{\frac{1}{m}\sum_{k\in S_{i}}{\left(x_{k}-\mu_{i}\right)}^{2}+\omega }$$
where ω is a very small constant introduced to prevent the denominator in Equation (2) from becoming zero, $S_{i}$ is a pixel set, and m is the number of pixels.

For BN, its pixel set depends on a, which is defined as shown in (4):(4)$$S_{i}=\{k|k_{C}=i_{C}\}$$
as can be seen from Formula (4), BN normalizes each channel and calculates the mean and standard deviation along (N,H,W).

For GN, its pixel set depends on N, which is defined as shown in (5):(5)$$S_{i}=\{k|k_{N}=i_{N}\}$$Unlike BN, GN calculates the mean and standard deviation along (C,H,W), which normalizes each batch and directly avoids the influence of batch size. Using Conv_GU as the basic module of LSCHead can significantly enhance the classification and regression performance of the detection head. At the same time, the idea of sharing parameters also makes it easier to deploy the model on edge devices. Considering the problem of different feature scales, this paper further selects the Scale layer to scale the features.

## 4. Experiment and Evaluation

### 4.1. Experimental Conditions

The experimental setup for defect detection, as detailed in this paper, comprises an NVIDIA GeForce RT 4090 24GB GPU, with the operating system running on Ubuntu 20.04.4. The compilation environment includes Python version 3.8.18, the torch framework version 1.13.1, and CUDA version 11.6. The baseline model is YOLOv8n. 

### 4.2. Dataset Introduction

The dataset used in this experiment is a data sample of substation equipment from the State Grid Shandong Electric Power Company, which includes 12,968 data. There are 12 types of defects in total. In the experiment, the dataset is shuffled and re-divided, the overall dataset is partitioned into a training set, a test set, and a validation set, with a ratio of 8:1:1. The specific number is 10,374 training sets, 1297 test sets, and 1297 validation sets. The specific data sample example is shown in Figure 7.

### 4.3. Evaluation Indicators

Considering the detection accuracy required for substation defect detection and the characteristics of easy deployment on mobile devices, this paper employs mean Average Precision (mAP) as a metric to assess the accuracy of substation defect detection and utilizes parameters and GFLOPs to evaluate the complexity of the proposed model. The quantity of model parameters and the magnitude of the FLOPs directly impact the complexity of deploying the model. To more rigorously validate the outcomes, this paper employs two metrics, mAP50 and mAP50-95, for model evaluation. The computation of mAP necessitates the determination of Precision (*P*) and Recall (*R*). Precision signifies the model’s rate of detecting defects, while Recall indicates the model’s ability to identify all defective instances. The calculation formula is as follows:(6)$$P=\frac{TP}{TP+FP}$$
(7)$$R=\frac{TP}{TP+FN}$$

Within this context, True Positive (*TP*) denotes the quantity of positive instances accurately recognized by the classifier. False Positive (*FP*) refers to the instances incorrectly labeled as positive. Conversely, False Negative (*FN*) indicates the instances incorrectly classified as negative. Average Precision (*AP*) serves to measure the precision of a specific category label, and its computation is performed using the following formula:(8)$$AP=\int_{0}^{1}P(R)dR$$
the PR curve is obtained by Formulas (1) and (2).

mAP represents the accuracy across all category labels. The greater the mAP, the superior the model’s overall performance. The calculation formula is as follows:(9)$$mAP=\frac{\sum_{i=1}^{k}AP_{i}}{k}$$
where *k* denotes the total number of categories.

The calculation of mAP has different measurement standards based on different Intersection Over Union (IOU). IOU represents the overlap between the detection box predicted by the model and the real annotation box. When the detected overlap is greater than a preset value, it is judged to be correct. mAP50 and mAP50-95 are different mAP values obtained when the preset values are 50% and 50–95%. The detection accuracy in this article is based on mAP50.

The quantity of model parameters and the volume of FLOPs directly influence the complexity of deploying the model on mobile devices. The more complex the model, the larger the total number of parameters. The FLOPs represents the amount of floating-point operations in the model. The larger the amount of FLOPs, the slower the inference verification of the model, and the more difficult it is to deploy on devices with poor computing power.

### 4.4. Parameter Settings

Regarding parameter configuration, all hyperparameters in this experiment have been optimized, and the entire model has been trained for 500 epochs to ensure its convergence. To streamline the training process, mosaic data augmentation is disabled during the final 10 epochs. The image input is consistently set to 640 × 640, the batch size is established at 16, and the optimizer is configured as Stochastic Gradient Descent (SGD). Ensuring that subsequent experiments are conducted under this optimized parameter setup.

As shown in Figure 8, the loss comparison between YOLOv8n and EAL-YOLO under this parameter setting. Upon examining the comparative chart, it becomes evident that within the initial 100 training rounds, both algorithms experience a rapid decline in loss. Subsequently, between the 100–500 rounds of training, the model appears to approach convergence. Compared with YOLOv8n, the loss of EAL-YOLO decreases more significantly during training, and it shows obvious advantages in the first 100 rounds of training.

### 4.5. Comparative Experiment on Small Target Defect Detection

Among the defects of substation equipment, small target defect detection has always been the focus and difficulty. During detection, the small size of minor target defects makes them difficult to identify, often resulting in missed and false detections. The EAL-YOLO proposed in this paper is optimized for small target defects. In order to more intuitively see the advantages of EAL-YOLO in small target defect detection, this paper uses heatmap to analyze the model, selects HiResClassActivationMapping (HiResCAM) [28] as the heat map generation method, and uses the P2-P5 four-layer features of the model as input. The output heatmap is shown in Figure 9. Figure 9 shows the heatmap analysis of YOLOv8n and EAL-YOLO in the same scene, where the first line is YOLOv8n and the second line is EAL-YOLO. Among them, the brighter the area in the figure, the more attention the model pays. As can be seen from Figure 9, both YOLOv8n and EAL-YOLO can get the attention of larger defect targets, but from the perspective of the highlighted area, YOLOv8n’s attention is more scattered, and EAL-YOLO’s attention is more concentrated. For small target defects, as depicted within the green box region in Figure 9, YOLOv8n in Figure 9 is not sensitive to small target defects, which will lead to problems such as missed detection, while EAL-YOLO pays special attention to small target defects and can accurately detect small-size defect targets.

To more intuitively observe the model’s effectiveness in identifying small target defects, this paper divides and verifies the defect data verification set according to different detection frame sizes, and quantitatively compares the detection accuracy according to the three types of detection frame sizes, as shown in Table 1. As illustrated in Table 1, the detection frame size is divided into three sizes according to pixels: less than 100, 100 to 1000, and more than 1000. Among them, the size below 100 is a small target. From the data in the table, it is evident that for small targets with dimensions below 100, EAL-YOLO has significantly improved the effect compared with the original YOLOv8n, and has increased by 5.3 percentage points in mAP50. EAL-YOLO also has significant advantages on medium-sized defect targets between 100 and 1000 and large targets of more than 1000.

### 4.6. Comparison of Typical Defect Detection Results

In order to verify the effectiveness of the EAL-YOLO, this paper detects 12 types of typical defects, and the detection scenarios cover indoor defects and outdoor defects. The accuracy is shown in Table 2. The highest accuracy can reach 97.6%, and the detection accuracy of 12 types of defects can reach more than 85%. Among them, EAL-YOLO is better than YOLOv8n in accuracy, especially in small target defect types. For example, when the indicator light is abnormal, its accuracy is improved by 9.7 percentage points in the light-on state and 5.9 percentage points in the light-off state.

Figure 10 illustrates the outcomes of defect detection using YOLOv8n and EAL-YOLO. The selected images are all verification images that did not participate in the training. The first row is the detection effect of YOLOv8n, and the second row is the detection effect of EAL-YOLO. In the displayed detection results, the boxed labels are the detected equipment and defects. Only the defects are analyzed here. As shown in Figure 10a, in a scene with many small targets, YOLOv8n has a poor detection effect on small targets, and the indicator light and pressure plate status are missed, while EAL-YOLO detects the indicator light status and pressure plate status that YOLOv8n does not detect. In Figure 10b, for the defect detection of the breather, YOLOv8n only detects the discoloration defect of the breather silicone, while EAL-YOLO detects the oil seal damage defect that YOLOv8n does not detect. In Figure 10c, in a scenario with many defects, YOLOv8n only detected the discoloration defect of the breather silicone, and the abnormal oil seal defect was misdetected as a normal oil seal, while EAL-YOLO accurately identified the abnormal oil seal defect and detected the abnormal oil level defect of the gas observation window that YOLOv8n did not detect. In Figure 10d, in a complex scenario, YOLOv8n did not detect the defect, but EAL-YOLO accurately identified the damaged dial defect. It can be clearly seen from Figure 10 that in the substation scenario, EAL-YOLO can effectively detect defect information in small targets and complex backgrounds, and overcome the false detection and missed detection phenomenon of YOLOv8n.

### 4.7. Ablation Experiment and Comparative Experiment

To ascertain the efficacy of the enhancements suggested in this paper, Table 3 presents the outcomes of the ablation study. The data are clearly visible within Table 3. After Algorithm 1 introduced EfficientFormerV2 as the network backbone, the model’s capability for multi-scale feature extraction has been enhanced, and the accuracy of the model was enhanced by 0.4 percentage points. To enhance network performance more effectively, Algorithm 2 added the LSKA mechanism on the basis of Algorithm 1. Compared with Algorithm 1, the accuracy of Algorithm 2 was improved by 0.6 percentage points. Algorithm 3 introduced ASF2-Neck on the basis of Algorithm 2. Compared with Algorithm 2, the accuracy of Algorithm 3 was improved by 0.68 percentage points. Algorithm 4 introduced LSCHead, which shares convolution information, for model lightweighting. Compared with Algorithm 3, the accuracy of Algorithm 4 increased by 1.25 percentage points, while the number of parameters decreased by 14%, the amount of FLOPs decreased by 21.9%, and the difficulty of model deployment was reduced. In summary, Algorithm 1 and Algorithm 2 mainly enhance the model’s feature extraction capabilities and improve the model’s accuracy. Algorithm 3 mainly enhances the model’s small target defect detection capabilities and further improves the model’s accuracy. Algorithm 4 explores the lightweighting of the model based on the first three, achieving the lightweighting of the model while ensuring accuracy.

Table 4 presents a comparative analysis of EAL-YOLO against other models. The compared models are roughly divided into YOLO series and non-YOLO series. Among the YOLO series algorithms, several typical algorithms of YOLOv3-v8 are selected. Among the non-YOLO series, the two-stage algorithm Faster-RCNN and the one-stage algorithms RetinaNet [27], ATSS [29] and TOOD [30] are selected. These algorithms are all representative algorithms in target detection. It is evident that among the various target detection models evaluated, the YOLOv8 model demonstrates superior performance. At the same time, the EAL-YOLO proposed in this paper improves mAP50 by 2.93 percentage points compared with the original YOLOv8n. Simultaneously, the quantity of parameters and the volume of FLOPs have been diminished by 61.17% and 46.5%, respectively, in comparison to YOLOv8s.

## 5. Conclusions

In view of the complex scenes of substations and the large span of defect sizes, this paper proposes a small target and lightweight substation equipment defect detection algorithm, EAL-YOLO, which enhances the precision of defect detection while considering the intricacies of the model. From the experimental results, EAL-YOLO has the following advantages over YOLOv8: 

(1) Stronger feature extraction capabilities. When facing complex scenes, EAL-YOLO can effectively extract scene feature information and detect the defect types in complex scenes, effectively avoiding the phenomenon of false detection and missed detection by YOLOv8n. Compared with YOLOv8n, EAL-YOLO has enhanced its detection accuracy by 2.93 percentage points.

(2) More accurate small target defect detection capability. EAL-YOLO is better than YOLOv8n in paying attention to small target defects and can detect smaller defect types. Compared with YOLOv8n, EAL-YOLO has improved its accuracy by 5.3 percentage points in small target defect detection with a size of less than 100 pixels.

(3) Lighter model architecture. Compared with YOLOv8s, the detection accuracy of EAL-YOLO has been improved while the number of parameters and the amount of FLOPs have been reduced by 61.17% and 46.5% respectively, meeting the needs of lightweight models. 

(4) In actual detection, facing 12 typical defect types, EAL-YOLO can meet its mAP50 of 92.26%, among which the highest defect type AP50 is 97.6%. In general, the EAL-YOLO proposed in this paper meets the needs of substation defect detection.

## Figures and Tables

**Figure 1 sensors-24-05914-f001:**
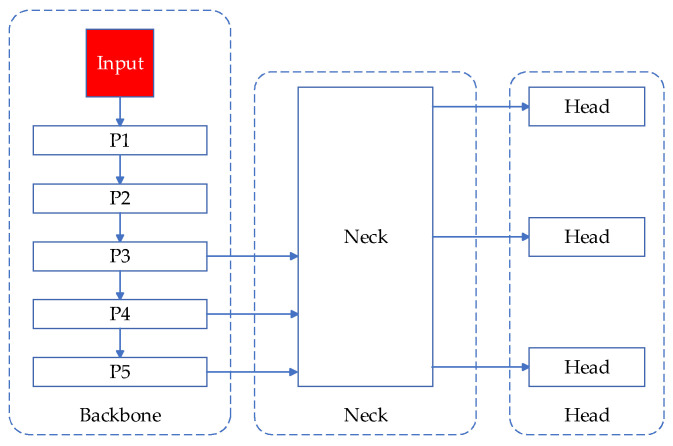
YOLOv8 model structure.

**Figure 2 sensors-24-05914-f002:**
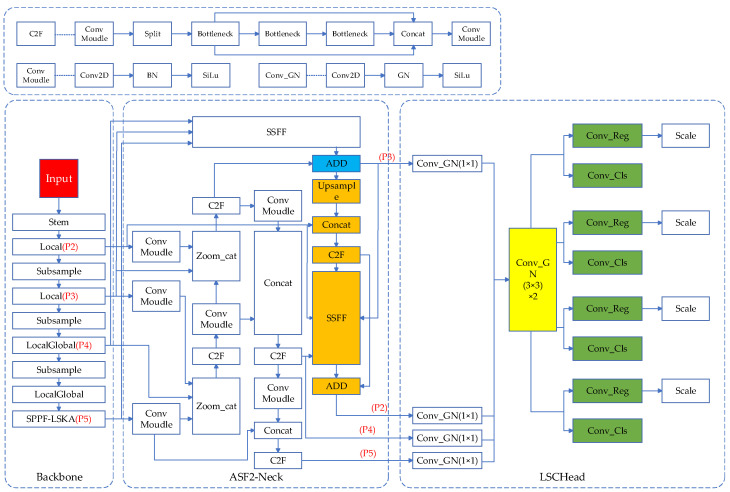
EAL-YOLO model structure.

**Figure 3 sensors-24-05914-f003:**
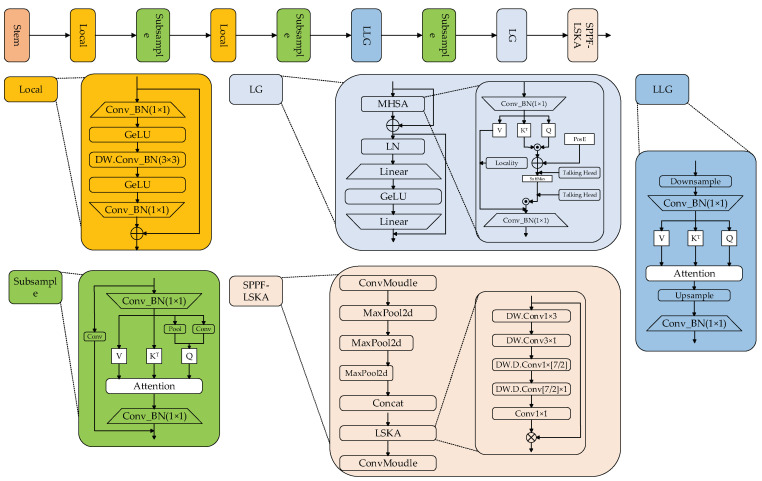
EAL-YOLO backbone structure.

**Figure 4 sensors-24-05914-f004:**
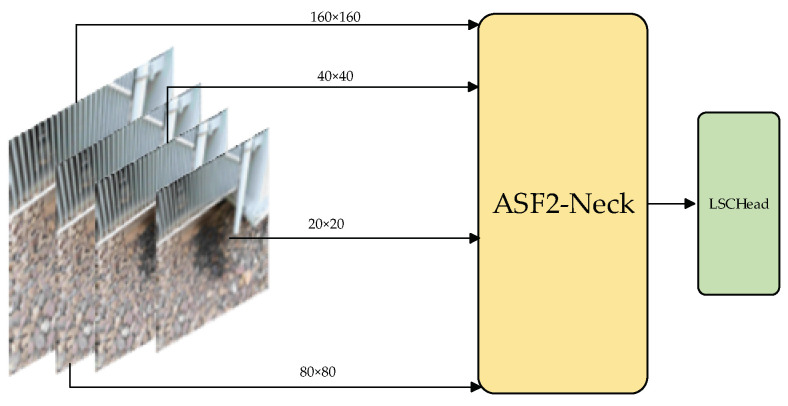
Model architecture after adding P2 feature output layer.

**Figure 5 sensors-24-05914-f005:**
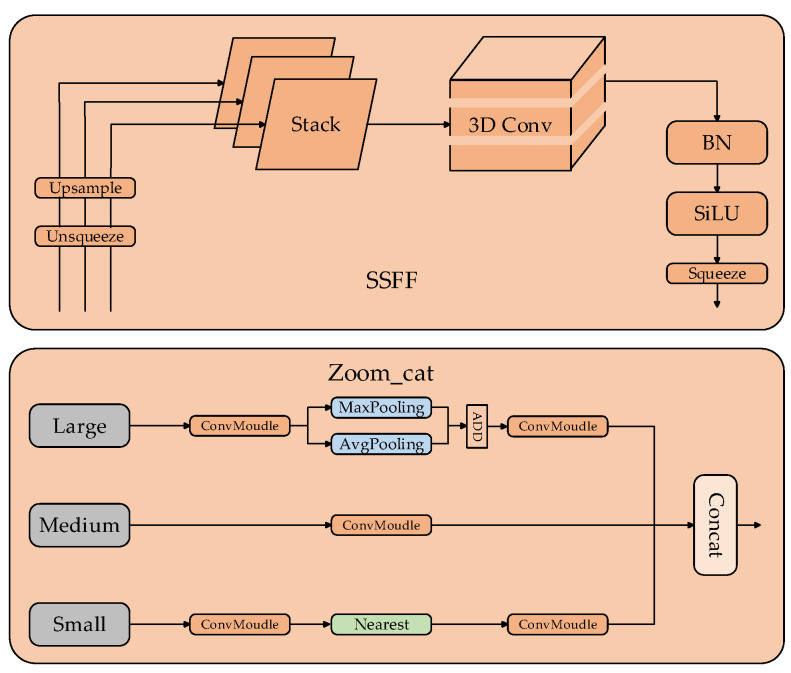
SSFF and Zoom_cat structure.

**Figure 6 sensors-24-05914-f006:**
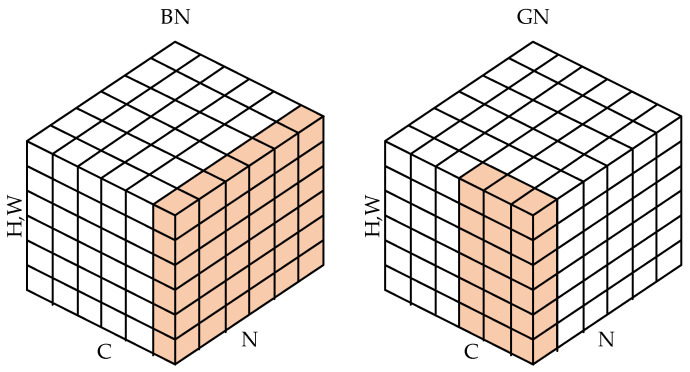
BN and GN structures.

**Figure 7 sensors-24-05914-f007:**
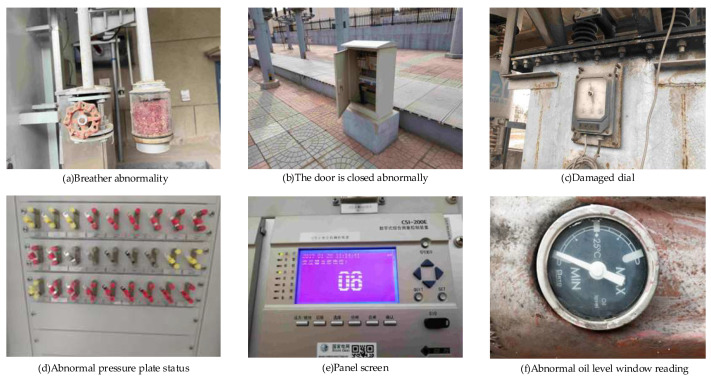
Substation equipment defect sample example.

**Figure 8 sensors-24-05914-f008:**
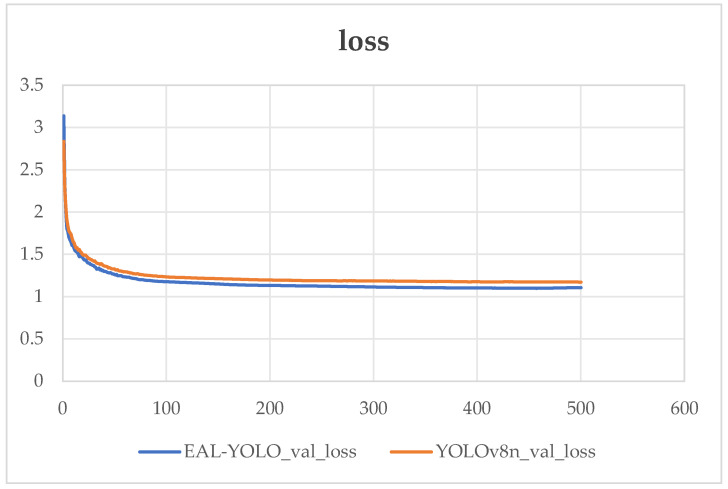
Comparison between YOLOv8n and EAL-YOLO loss.

**Figure 9 sensors-24-05914-f009:**
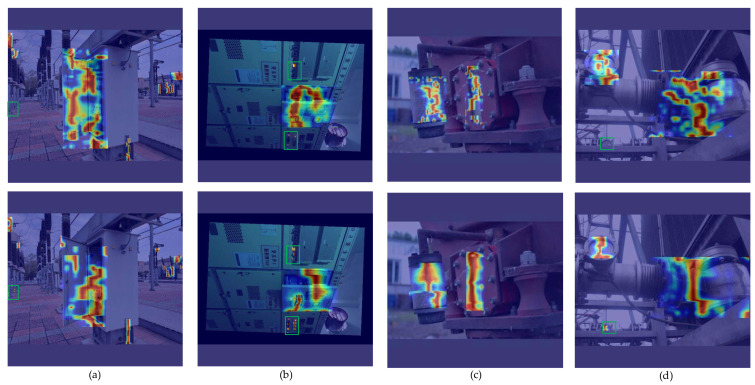
Heatmap analysis of YOLOv8n and EAL-YOLO: (**a**) The main defect type is the abnormal door closure defect; the green box indicates whether the small target defect type is detected; (**b**) a secondary equipment type defect; the abnormal door closure is detected. The green box indicates the abnormal indicator light status defect; (**c**) the discoloration of the respirator silicone and the abnormal oil level window reading; (**d**) a meter defect type; the green box indicates whether the small target meter defect is detected.

**Figure 10 sensors-24-05914-f010:**
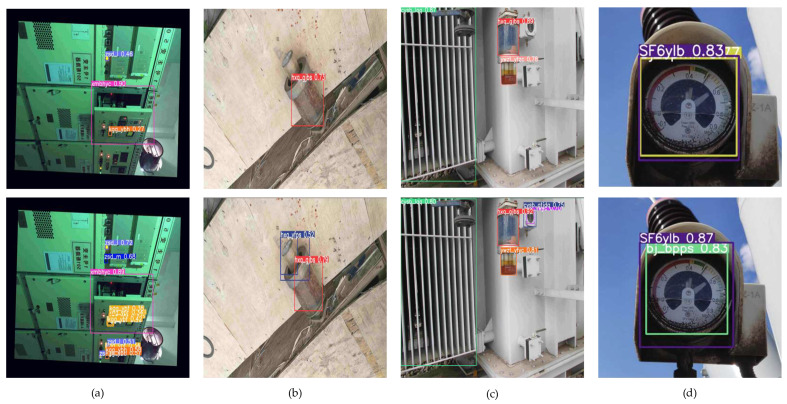
The detection results of YOLOv8n and EAL-YOLO (boxes of different colors represent different detection categories): (**a**) the door closing defect and the status of several indicator lights and the pressure plate; (**b**) the breather silicone discoloration defect and the oil seal damage defect; (**c**) the breather silicone discoloration defect, the oil seal abnormal defect, and the gas observation window oil level abnormal defect; (**d**) the dial damage defect.

**Table 1 sensors-24-05914-t001:** Comparison of detection capabilities of data models of different sizes.

Data Size (Pixels)	YOLOv8n		EAL-YOLO	
	AP50	AP50-95	AP50	AP50-95
(~,100)	85.5%	57.03%	**90.8%**	**63.4%**
[100,1000)	92.43%	78.31%	**94%**	**79.5%**
[1000,~)	82.9%	58.1%	**85.9%**	**59.4%**

**Table 2 sensors-24-05914-t002:** Comparison of accuracy of typical defect types.

Typical Defect Types		Label Name	YOLOv8n		EAL-YOLO	
			AP50	AP50-95	AP50	AP50-95
Breather silicone broken		hxq_gjps	89%	72.9%	**90.7%**	**75.2%**
Abnormal pointer reading		bjdsyc_zz	93.3%	78.1%	**93.6%**	**79.6%**
The door is closed abnormally		xmbhyc	82.9%	58.1%	**85.9%**	**59.4%**
Breather oil seal damaged		hxq_yfps	85.9%	67.1%	**91.3%**	**71.7%**
Panel screen		mbhp	96.7%	90.9%	**94.9%**	**93.3%**
Abnormal oil level window reading		bjdsyc_ywc	92.8%	80.2%	**95.3%**	**78.4**
Damaged dial		bj_bpps	92.1%	81.7%	**94.6%**	**81.8%**
The silicone barrel of the breather is damaged		hxq_gjtps	97.2%	77.3%	**97.6%**	**76.5%**
The switch cabinet pressure plate is in abnormal state	close	kgg_ybh	89.8%	67.3%	**89.6%**	**69.4%**
	open	kgg_ybf	88.8%	69.8%	**94.6%**	**76.2%**
The indicator light is abnormal	bright	zsd_l	80.8%	47.4%	**90.5%**	**56.2%**
	dark	zsd_m	82.6%	43.6%	**88.5%**	**51.8%**

**Table 3 sensors-24-05914-t003:** Ablation experiment.

Model	EfficientFormerV2	LSKA	ASF2- Neck	LSCHead	mAP50	mAP50-95	Params/M	GFLOPs
YOLOv8n	/	/	/	/	89.33%	69.53%	3.09	8.5
Algorithm 1	√	/	/	/	89.73%	70.38%	5.19	12.1
Algorithm 2	√	√	/	/	90.33%	70.86%	5.32	12.2
Algorithm 3	√	√	√	/	91.01%	71.94%	5.04	19.6
Algorithm 4	√	√	√	√	92.26%	72.46%	4.33	15.3

**Table 4 sensors-24-05914-t004:** Comparative experiments.

Model	mAP50	mAP50-95	Params/M	GFLOPs
RetinaNet	82.9%	63.9%	36.517	210
YOLOv3 [31]	79.35%	62.43%	12.17	19
YOLOv5	89.13%	69.34%	2.59	7.5
YOLOv6 [32]	87.83%	68.74%	4.45	12.8
YOLOv8n	89.33%	69.53%	3.09	8.5
YOLOv8s	91.73%	72.39%	11.15	28.6
Faster-RCNN	83.7%	63.5%	41.39	208
ATSS	79.4%	61.7%	38.91	110
TOOD	82.9%	65.1%	32.04	199
EAL-YOLO (Ours)	**92.26%**	**72.46%**	**4.33**	**15.3**

## Data Availability

The datasets presented in this article are not readily available because [the data in this article belongs to State Grid Shandong Electric Power Research Institute. Used in the Science and Technology project of State Grid Shandong Electric Power Company. Due to the confidentiality of the project and other reasons, it is not possible to directly access the datasets]. Requests to access the datasets should be dircted to [corresponding author: lysgwork@163.com].

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
