# Peer review of "Lightweight Substation Equipment Defect Detection Algorithm for Small Targets"

_sensors, 2024, doi:10.3390/s24185914_

Round 1

Reviewer 1 Report

Comments and Suggestions for Authors

This manuscript is well-written. I have only some minor suggestions.

1.The authors should explain the steps taken to obtain the heatmaps.

2. Maintain consistency in the font used in the figures, such as in Figure 4.

3. Briefly explain why you have chosen these methods as comparison methods, such as ATSS, RetinaNet, etc.

4. In the conclusion section, it is recommended to present the findings in the form of bullet points.

Reviewer 2 Report

Comments and Suggestions for Authors

Before being accepted, please revise this paper in accordance with the following review comments:

1.     Authors are requested to give the full name of the method the first time the acronym is used, such as the meaning of EAL in EAL-Yolo v8.

2.     The authors of the introduction section are missing literature on the use of deep learning models for related tasks in the same field.

3.     In the ablation experiments the authors should indicate which of the four algorithms brought about the increase in accuracy.

4.     There are several competing intelligent detection methods that are missed, that should have been cited and referred to. The following are just some examples: “A Novel Fault Detection Model Based on Vector Quantization Sparse Autoencoder for Nonlinear Complex Systems”, “A multi-source domain information fusion network for rotating machinery fault diagnosis under variable operating conditions”.

Round 2

Reviewer 2 Report

Comments and Suggestions for Authors

No more comments.